chemical biology/biomathematics

scanning electron microscopy, hair damage, image analysis, microscopic hair analysis, surface roughness

**Author for correspondence:**
Fanny Chu
e-mail: chu28@llnl.gov

This article has been edited by the Royal Society of Chemistry, including the commissioning, peer review process and editorial aspects up to the point of acceptance.

# Automated analysis of scanning electron microscopic images for assessment of hair surface damage

Fanny Chu[1,2], Deon S. Anex[1], A. Daniel Jones[3] and Bradley R. Hart[1]

[1]Lawrence Livermore National Laboratory, 7000 East Ave., Livermore, CA 94550, USA
[2]Department of Chemistry, Michigan State University, 578 S Shaw Ln, East Lansing, MI 48824, USA
[3]Department of Biochemistry and Molecular Biology, Michigan State University, 603 Wilson Road, East Lansing, MI 48824, USA

FC, 0000-0002-1114-6182; DSA, 0000-0002-3491-9379; ADJ, 0000-0002-7408-6690; BRH, 0000-0002-2832-4168

Mechanical damage of hair can serve as an indicator of health status and its assessment relies on the measurement of morphological features via microscopic analysis, yet few studies have categorized the extent of damage sustained, and instead have depended on qualitative profiling based on the presence or absence of specific features. We describe the development and application of a novel quantitative measure for scoring hair surface damage in scanning electron microscopic (SEM) images without predefined features, and automation of image analysis for characterization of morphological hair damage after exposure to an explosive blast. Application of an automated normalization procedure for SEM images revealed features indicative of contact with materials in an explosive device and characteristic of heat damage, though many were similar to features from physical and chemical weathering. Assessment of hair damage with tailing factor, a measure of asymmetry in pixel brightness histograms and proxy for surface roughness, yielded 81% classification accuracy to an existing damage classification system, indicating good agreement between the two metrics. Further ability of the tailing factor to score features of hair damage reflecting explosion conditions demonstrates the broad applicability of the metric to assess damage to hairs containing a diverse set of morphological features.

# 1. Introduction

Microscopic analysis of hair finds utility in diverse disciplines, such as in medical and forensic sciences, though confidence in the interpretations of this type of analysis varies widely. In the context of forensic science, microscopic hair analysis, a qualitative approach, has shown limited discriminative power [1,2], ranging from use as a predictor of short-tandem repeat typing success from DNA in hair [3,4] to comparative microscopy to aid forensic identification. Clinically, microscopic analysis can be used as a tool to assess hair damage, as an indicator of health status [5–7], but analyses are performed in a qualitative manner through identification of morphological features of hair damage. Exposure to various physical and chemical stresses including detergents, dyes, brushing, and UV light alters hair structure [8–11], and is of considerable interest for diagnosis of dermatological conditions; however, few studies have quantified the extent of hair damage based on morphological features [12–14]. Notably, Kim *et al.* developed a classification system with five damage grades for characterizing hair surface damage from weathering [12], which was then expanded upon to a 12-point scale by Lee and colleagues [14], though the grading systems were dependent on visual scoring of scanning electron microscopic (SEM) images based on subjective evaluations of severity in the irregularity of hair cuticular structure. Microscopic analysis remains a predominantly qualitative technique via visual assessments; little emphasis has been placed on developing more objective metrics and even less so on quantitation of hair damage severity.

Digital image analysis has been underutilized for classification of hair fibres from various microscopic methods, despite offering potential for more objective detection and comparison of image features. Of these studies, the majority concentrated on morphological features detected by light microscopy and analysed using commercial software [15–19], even though other microscopic techniques such as atomic force microscopy (AFM) and SEM permit more extensive hair structure analysis for comparison at higher spatial resolution [20,21]. In particular, Gurden *et al.* assessed hair structural damage from bleaching, and differentiated root and distal ends affected by chemical treatment with cuticular structure measurements such as surface roughness from AFM images [20], though the extent of damage was not graded or quantified. There is a need for development of an objective scoring system to characterize the extent of hair damage using automated analysis of higher resolution microscopic images.

However, hair-to-hair variation and image acquisition differences obfuscate characterization and scoring of hair surface damage from microscopic images, which rely on feature detection and pairwise comparisons. Structural differences between two hair segments (e.g. width, curvature), even along the length of a hair, and automatic setting of brightness and contrast parameters for optimal SEM image acquisition make feature detection and hair segment comparison in image analysis challenging. Briefly, SEM imaging to interrogate specimen surface topography is achieved as an electron beam scans over the specimen via interactions between primary electrons and accessible atoms from the specimen, leading to the emission of secondary electrons [22]. The number of secondary electrons that reach the detector manifests as pixel greyscale brightness in an image [23]. Pixel brightness contrast within an image provides topographical information about the specimen surface, which is most accessible to the primary electrons, and is affected by, among other factors, the ease with which secondary electrons escape the surface of the specimen once formed [22]. However, detection of morphological features on the surface for image comparison may be complicated when brightness contrast varies within and between images. For example, SEM images of segments from two different hairs (Hair Samples 4 and 2), shown in figure 1*a* and 1*b*, respectively, can display vastly different brightness levels, even within the same image along the width of the hair segment, owing to hair fibre positions and stage tilt angles. Furthermore, the tubular structure of hair creates different contact angles for the electron beam, which affects formation and detection of secondary electrons. Coupled with the orientation of the electron beam and the detector with respect to the hair fibre, detection of an abundance of secondary electrons formed from contact with hair segment edges manifests as abnormally bright bands along the edges of the hair segment that obscure image features entirely, even after carbon coating. To facilitate feature identification and enable direct comparison of different images, such artefacts must be removed or addressed.

While many normalization methods have been implemented to remove image artefacts such as brightness variation, procedures used to process digital images focus on contrast enhancement. These include variations of histogram equalization, gamma intensity correction (GIC), and wavelet-based methods for applications such as feature detection in retinal and magnetic resonance images for

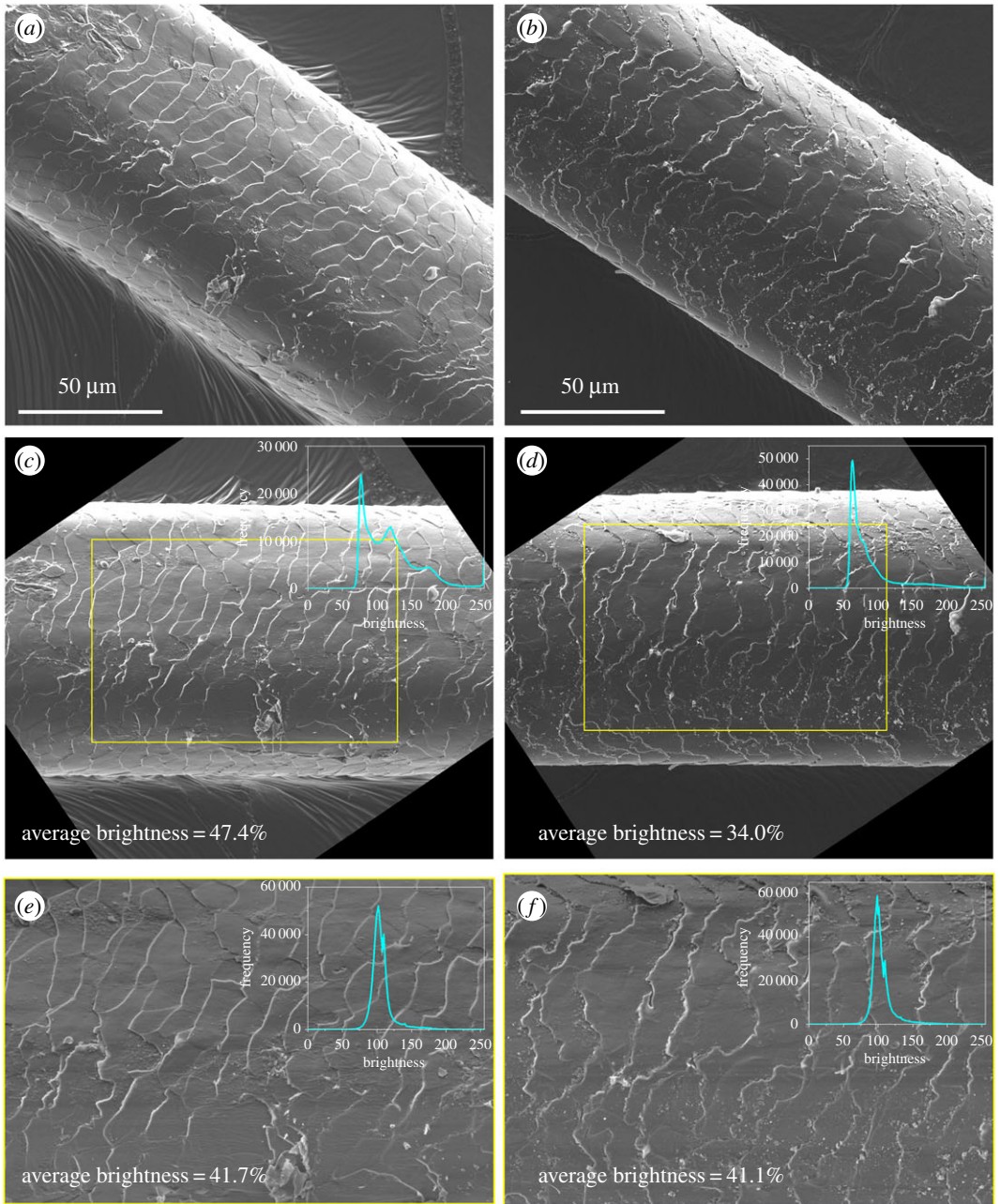

**Figure 1.** SEM images of (*a*) and (*b*) hair segments as original raw images, (*c*) and (*d*) rotated segments with selected region of interest (yellow rectangle) and image brightness histogram of region, and (*e*) and (*f*) regions of interest after normalization and corresponding image brightness histograms, from Hair Samples 4 and 2, respectively. Original images (*c*) and (*d*) show different brightness levels due to hair-to-hair variation and image acquisition differences. The described normalization procedure minimized brightness differences within and between images, as displayed in (*e*) and (*f*).

disease diagnosis [24,25] and in digital images for facial recognition [26,27]. However, these methods necessitate user inputs and parameter optimization, such as the gamma value in GIC, and are used to enhance features for detection in an image. Variations in both parameters are not conducive to comparative image analysis with a scoring system. Instead, desirable normalization procedures require minimal user-defined inputs, are computationally inexpensive, and preserve pixel brightness information in an image while reducing hair-to-hair and image acquisition variation.

We aimed to facilitate identification of microscopic features characteristic of hair damage and image comparison by developing and applying a simple and automated normalization procedure and evaluate metrics for representing hair damage. Development and automation of image analysis for assessment of hair surface damage from this preliminary study directly enables correlations of the effects of an explosive blast on morphological hair damage with alterations in chemical composition of hair, which

will be further explored in a future publication. Using open source image visualization program ImageJ [28,29], morphological features unique to hairs subjected to explosive blast conditions were identified after normalization. Morphological features of damage exhibited higher pixel brightness owing to their elevation or depression from the hair cuticle surface, which can be detected as peak lag tailing in pixel brightness histograms. Metrics to quantitate pixel brightness from these features, including roughness and tailing factor, were evaluated for broadly applicable scoring of hair surface damage.

# 2. Methods

## 2.1. Hair sample collection

Scalp hair specimens were collected as part of a larger study to identify peptide biomarkers for protein-based human identification. Hair fibres from two individuals were used in this study; see Section End Statements for more details on approval for hair sample collection. After assembly of an experimental explosive device using commercial materials as part of a training exercise hosted by the Bureau of Alcohol, Tobacco, Firearms, and Explosives National Center for Explosives Training and Research at the Redstone Arsenal in Huntsville, AL, hairs (less than 5 cm) were taped onto the internal and external regions of the device. The device was then detonated in a spherical total containment vessel with a diameter of 48 in (1.2 m) using 2 inches (5 cm) of dynamite. Remnants of the device and hair fibres were collected from the total containment vessel after the explosion and stored in the dark at room temperature.

## 2.2. Hair sample preparation for scanning electron microscopy image acquisition

Recovered hair fibres were isolated and segmented; one inch (approx. 2.5 cm) each was allotted for protein analysis that will be described in a future publication, and the remainder (approx. 1 cm) was used for scanning electron microscopic (SEM) analysis. Three exploded and two control hairs were randomly selected for analysis and image acquisition. Each segmented hair was fixed onto a stub prior to carbon coating; under vacuum, a carbon layer of approximately 10 nm was deposited onto each specimen after heating for approximately 5 s. Secondary electron images were acquired along the length of each hair fibre using an Inspect F (FEI Company, Hillsboro, OR) scanning electron microscope, at an acceleration voltage of 5 kV, a dwell time of 3 µs, and a working distance of 7 mm over a range of magnifications. Brightness and contrast were automatically adjusted for each image. In total, 58 digital SEM images (8-bit) were acquired from five hair segments, and all were then processed using ImageJ 1.52k software in replicates of $n = 5$.

## 2.3. Automated image normalization procedure

Prior to normalization, a region of interest (ROI) was computationally defined in each image to ensure that regions exhibiting abnormally high brightness on the edges of hair segments were excluded from the area processed by image analysis. The original raw image was first rotated using a user-defined line input along the length of the hair segment so that the length of the hair segment was oriented along the horizontal axis of the ROI. Empirical evaluations of a few hair segment images showed that up to 10 µm of hair surface along the vertical axis from either edge were prone to abnormally high brightness, approximately 20% of the width of hair segment. To uniformly define the ROI bounds between images yet exclude abnormally bright regions, 75% of the hair segment width and length equidistant from the image centre were included in the ROI. The bounds, length, and width of the hair segment were then defined (in pixels) using a user-defined diagonal line, with coordinates $(x_1, y_1)$ and $(x_2, y_2)$, that spanned two corners of the segment. From the diagonal line, the upper left-hand corner coordinates $(x_{lh}, y_{lh})$, length, and width of the ROI were defined according to equations (2.1)–(2.3):

$$(x_{lh}, y_{lh}) = \left( \frac{7x_{min} + x_{max}}{8}, \frac{7y_{min} + y_{max}}{8} \right), \tag{2.1}$$

$$l = \frac{3}{4}(x_{max} - x_{min}), \tag{2.2}$$

and

$$w = \frac{3}{4}(y_{max} - y_{min}), \tag{2.3}$$

where $x_{min}$, $x_{max}$, $y_{min}$, and $y_{max}$ represent the minima and maxima of $x$ and $y$, respectively, extracted from the diagonal line described by coordinates $(x_1, y_1)$ and $(x_2, y_2)$. Centring the ROI to encompass

9/16th (or 56%) of the hair segment area ($l \times w$) allows most of the segment to be included for image analysis while excluding edge regions where features are entirely obscured due to abnormal pixel brightness for reasons discussed above (figure 1c,d).

After ROI definition, brightness within an image was equalized by normalizing to the average brightness per row of pixels followed by centring the average at a value of 109 and rescaling. The value of 109 was selected empirically by considering two hair segment images and calculating the average image brightness within the ROIs from the two images and then averaging the obtained results. The resultant centring value of 109 is equivalent to 43% of the maximum brightness (from a scale ranging between 0 and 255) and represents a dark grey pixel. This pixel brightness value corresponds to undamaged regions of hair segments, which make up the majority of the pixels in the image. To preserve pixel brightness ratios with respect to the average brightness per row of pixels but ensure that average image brightness centres around 109 and pixel brightness maximizes at 255, normalization was performed for each pixel within the ROI according to equations (2.4) and (2.5):

$$I_{i,j,\text{norm}} = \frac{I_{i,j}}{\frac{1}{l}\sum_{n=1}^{l} I_{n,j}} \cdot 109 \tag{2.4}$$

and

$$I_{i,j,\text{norm,scale}} = \begin{cases} \dfrac{146 \cdot (I_{i,j,\text{norm}} - 109)}{I_{j,\text{norm,max}} - 109} + 109, & I_{i,j,\text{norm}} > 109 \\ I_{i,j,\text{norm}}, & I_{i,j,\text{norm}} \leq 109, \end{cases} \tag{2.5}$$

where $I_{i,j}$, $I_{i,j,\text{norm}}$, and $I_{i,j,\text{norm,scale}}$ represent the raw, normalized, and rescaled brightness of a pixel at image position $i,j$, respectively, $I_{n,j}$ is the brightness of a pixel at image position $n,j$ from 1 to ROI length $l$, $I_{j,\text{norm,max}}$ is defined as the maximum normalized brightness at the $j$th row, and 146 represents the difference between maximum pixel brightness 255 and brightness value 109. Equation (2.5) is based on min–max normalization, a common score normalization approach [30]. After normalization, ROIs exhibited less variance within an image and similar average brightness values (figure 1e,f).

## 2.4. Hair surface damage metric calculations

As morphological features of damage manifest as brighter pixels in contrast to undamaged regions, pixel brightness can be exploited for quantifying hair surface damage. The following metrics were investigated for scoring of hair surface damage in SEM images after normalization as different representations of pixel brightness: average image brightness, average image roughness, and tailing factor. Macros were written in ImageJ to carry out calculations for each metric on an image; see Section End Statements for code availability. Average image brightness $\overline{I_{\text{norm}}}$ of an ROI was calculated using the equation:

$$\overline{I_{\text{norm}}} = \frac{1}{l \cdot w} \sum_{j=1}^{w} \sum_{i=1}^{l} I_{i,j}, \tag{2.6}$$

where $I_{i,j}$ is the brightness of a pixel at image position $i,j$, and $l$ and $w$ represent the length and width of the ROI (in pixels), respectively.

Image roughness was evaluated for potential to quantify hair surface damage as an alternative metric of pixel brightness focusing on brightness fluctuation within an image to represent damage features. Based on the metric description by Gurden et al. [20], who previously reported use of roughness to profile hair cuticular surface, and adaptation of the distance formula for application to SEM images, average image roughness $\bar{r}$ was determined for $n$ sections along length $l$ of the ROI using equations (2.7)–(2.9):

$$s = \left\lfloor \frac{l}{n} \right\rfloor, \ 1 \leq s \leq l, \tag{2.7}$$

$$n_{\text{actual}} = \begin{cases} n, \left\{\dfrac{l}{s}\right\} = 0 \\ \lceil n \rceil, \left\{\dfrac{l}{s}\right\} > 0, \end{cases} \tag{2.8}$$

and

$$\bar{r} = \frac{1}{w} \sum_{j=1}^{w} \left[ \frac{1}{l} \left( \left( \sum_{i=1}^{n_{\text{actual}}-1} \sqrt{(I_{si+1,j} - I_{s \cdot (i-1)+1,j})^2 + (s)^2} \right) + \sqrt{(I_{l,j} - I_{s \cdot (n_{\text{actual}}-1)+1,j})^2 + (l - s(n_{\text{actual}} - 1))^2} \right) \right],$$

(2.9)

where $s$ is the section width, $n_{\text{actual}}$ is the total number of sections after accounting for dividends when sectioning $l$ by $s$ pixels, $w$ represents the width of the ROI, and $I_{si+1,j}$, $I_{s \cdot (i-1)+1,j}$, and $I_{s \cdot (n_{\text{actual}}-1)+1,j}$ are the pixel brightness values at image positions $si+1, j$, $s \cdot (i-1)+1, j$, and $s \cdot (n_{\text{actual}}-1)+1, j$, respectively, designated by the $i$th section.

A third metric, tailing factor, was examined for representing hair surface damage, since brighter morphological features of damage manifest as peak lag tailing in pixel brightness histograms. Tailing factor of an image brightness histogram, adapted from the USP measurement of chromatographic peak tailing [31], was determined in two steps: the peak apex was first redefined, bounded by the full width at half-maximum, to remove histogram skew created by the presence of multiple peaks, and then tailing factor was calculated at a fraction of the peak height maximum. Conventionally, the metric is calculated at 5% of the peak height maximum [31], though fraction $f$ was optimized between 1 and 10% of the peak height maximum for this application. Peak apex brightness $I_h$ and tailing factor $t_{fH}$ were determined using equations (2.10)–(2.12):

$$\overline{I_H} = \frac{\sum_{i=I_{\text{lead},0.5H}}^{I_{\text{lag},0.5H}} ic_i}{\sum_{i=I_{\text{lead},0.5H}}^{I_{\text{lag},0.5H}} c_i},$$

(2.10)

$$I_h = \begin{cases} I_H, & |\overline{I_H} - I_H| \leq 3 \\ \overline{I_H}, & |\overline{I_H} - I_H| > 3 \end{cases},$$

(2.11)

and

$$t_{fH} = \frac{I_{\text{lag},fH} - I_{\text{lead},fH}}{2(I_h - I_{\text{lead},fH})},$$

(2.12)

where $I_{\text{lead},0.5H}$ and $I_{\text{lag},0.5H}$ represent the peak lead and lag brightness, respectively, at 50% of the brightness profile peak height maximum $H$, $c_i$ is the frequency of pixel brightness $i$, $I_H$ is the brightness value at $H$, and $I_{\text{lead},fH}$ and $I_{\text{lag},fH}$ represent the peak lead and lag brightness at fraction $f$ of the peak height maximum.

## 2.5. Statistical analysis

All statistical analyses were performed in R (x64 version 3.4.4). Statistical significance was established at $\alpha = 0.05$. Pearson product-moment correlations of hair surface damage metrics followed by one-sample t-testing of the correlation coefficients were performed using the *cor.test* function in the *stats v3.5.3* package to determine statistical significance of the correlations. Training and test sets for a k-Nearest Neighbor Classification (kNN) model were established by randomization, each comprising 50% of the dataset and containing the same number of images from exploded and undamaged hairs. The model was developed using the *knn* function in the *class v7.3-15* package, with $k = 3$ nearest neighbours determined by Euclidean distance. All plots were drawn in OriginPro 2018 (OriginLab Corp., Northampton, MA).

# 3. Results and discussion

## 3.1. Identification of microscopic features for characterization of hair surface damage

Single hairs recovered after exposure to explosive blast conditions sustained damage comparable to that from physical and chemical weathering, as similar morphological features were identified in this study. Visual inspection of microscopic images of damaged hairs after normalization enabled identification of holes, cracks, lifting and tearing of the cuticle, and partial exposure of the cortex (figure 2a,b). Images were scored based on qualitative presence or absence of features, as described in the scanning electron microscopic (SEM) damage grade system proposed by Kim *et al.*, where overlapped cuticles represent the lowest degree of hair surface damage (Scu 1, or Grade 1 damage assessed in SEM images of hair

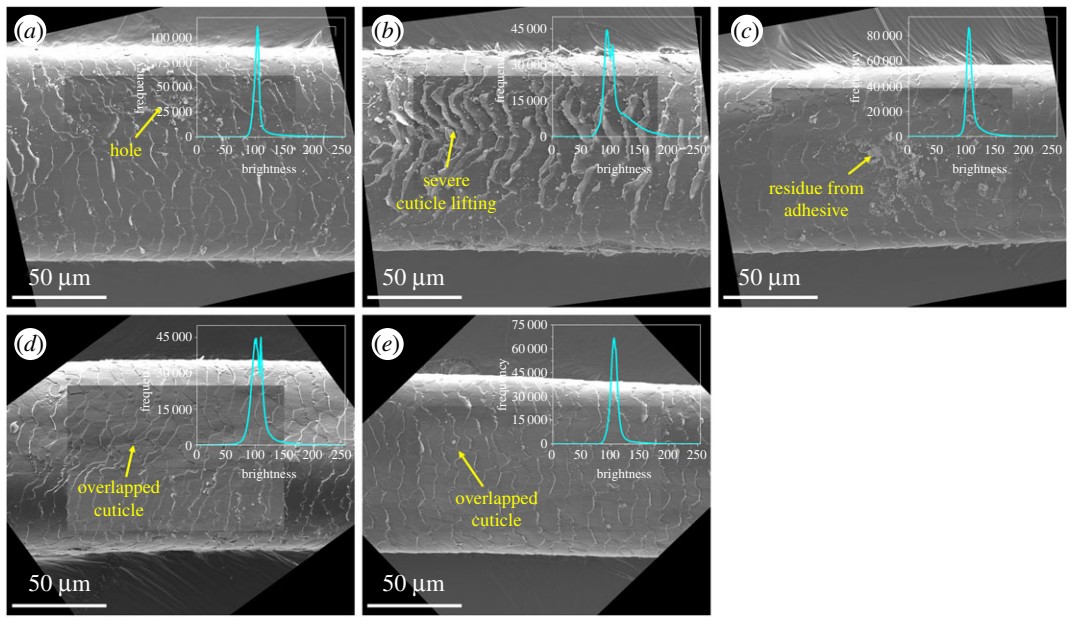

**Figure 2.** Representative rotated SEM images with overlays of normalized regions of interest and corresponding brightness histograms from Hair Samples 1–5, respectively. Features are labelled and denoted by yellow arrows. In addition to debris and particulates on the hair surface, features characteristic of damage from an explosion induced by an explosive device include (*a*) holes exposing layers of cuticle, (*b*) severe lifting of the cuticle edges and large cracks leading to partial exposure of cortex, and (*c*) localized non-specific cuticle lifting with residue from adhesive tape. Undamaged control hairs (*d*) and (*e*) predominantly display overlapped cuticles from daily weathering, illustrating substantially less severe hair surface damage compared to exploded hairs.

cuticle), apart from intact virgin hair, while the most severe hair damage (Scu 4) is characterized by the complete absence of cuticle and full exposure of cortex [12]. In particular, severe lifting of the cuticle edges in figure 2*b* most likely arose from scorching of the hair surface during the explosion, with the more intense scorching creating concavities in the edges of the cuticles along the centre of the hair fibre, as cuticle lifting is observed even when hair is exposed to 61°C temperatures from hair-drying [8]. Heat from the explosion also likely stressed the hair fibres, resulting in axial cracks on the surface along the hair length as the first indication of thermal damage, likely from cortical swelling in the fibre [8,32]. These features were also observed in hair exposed to physical and chemical weathering, such as frequent washing with detergent and exposures to bleach and UV light [12,13]. The extent of damage in recovered hairs varied along each hair. Regions in which damage consisted only of overlapped cuticles, attributed predominantly to daily weathering, were observed in exploded hairs, although the majority of SEM images containing this damage feature belonged to control hairs (figure 2*d*,*e*).

In addition to the above features, exploded hairs contained features not typically observed from physical and chemical weathering alone; embedded debris and particulates and cuticle lifting with adhered amorphous residue further characterized exploded hairs. Even without washing hair specimens after sample collection, control hair samples 4 and 5 were debris-free (figure 2*d*,*e*), indicating that the presence of embedded particulates is characteristic of physical contact with the explosive or remnants of the device. Furthermore, amorphous residue adhered to lifted cuticles (figure 2*c*) likely resulted during the hair fibre isolation process. Hairs previously attached to the experimental device via adhesive tape were isolated with forceps; detachment of hair fibres led to cuticle lifting, with residual adhesive bonded to the cuticle.

Normalization ensured that undamaged regions of hair fibres remain uniform in pixel brightness (intensity scale ranging 0–255) as a dark grey while physical features of hair surface damage appear as clusters of brighter pixels, ranging from light grey to white. For example, cuticle lifting is characterized by a cluster of brighter pixels, bounded by white pixels along the cuticle edges as distinct from the cuticle layer underneath, which is the result of elevation differences from the hair surface (figure 2*b*). In contrast, a depressed feature such as a hole manifests as alternating rings of light and dark pixels, as the light pixels delineate the edges of each exposed cuticle layer that is represented by darker pixels, down into the cortex (figure 2*a*). Similarly, many microscopic features of

hair surface damage identified herein are characterized by pixel brightness differences that can be further exploited in image analysis.

## 3.2. Evaluation of image parameters and metrics for scoring hair surface damage and image comparison

Automated image analysis for scoring hair surface damage and image comparison requires a reliable metric that not only represents the microscopic features identified above but can also be calculated from the image. As discussed above, many features characteristic of damage manifest as brighter pixels in images, compared to the uniform dark grey of smoother undamaged regions. Thus, we investigated the potential of using pixel brightness to score hair surface damage.

Although the simplest representation of brightness is its average, the metric was excluded from consideration due to its role in the normalization procedure and its correlation with magnification. The normalization procedure re-centred the average row brightness at a value of 109, which corresponds to 43% of the maximum brightness. This method not only readjusted brightness within images, but also effectively equalized brightness of undamaged regions between images for direct comparison. Therefore, average image brightness cannot be used to capture brightness differences between images from microscopic features of damage. Additionally, a significant positive correlation ($r = 0.595$; $p = 8.26 \times 10^{-7}$; figure 3$a$) between average image brightness and magnification indicates that images acquired at vastly different magnifications (between 1000× and 7000×) cannot be directly compared. Images exhibit greater average pixel brightness at higher magnifications, most likely from automatic brightness and contrast settings that were not modified when changing from low to high magnifications in the same region of the hair fibre, which result in higher overall brightness and low contrast for a smaller image area. Because of this correlation, only images acquired between 1000× and 4000× magnification were retained ($r = 0.059$; $p = 0.750$) for scoring and comparison.

We chose to explore metrics for assessing image roughness as an indication of hair surface damage. Physical surface roughness formed by elevations and depressions from the cuticle surface creates variations in adjacent pixel brightness that deviate from the average owing to differences in secondary electron trajectories from surface to detector. Roughness was previously used in conjunction with other metrics to profile morphological damage in the cuticular structure of human hair from images acquired via atomic force microscopy (AFM) in contact mode; using these metrics and multivariate statistics, Gurden *et al*. reported 86% accuracy to classifying hair segments as bleached versus untreated and from root or distal end [20]. Roughness was calculated from surface profiles over the length of the profile since AFM measures surface height, with a completely flat profile having a roughness defined as 1. But because height information is not directly obtained from SEM images, hair surface roughness determination was modified using the summation of pixel brightness differences between image sections over the length of the region of interest (equations (2.7)–(2.9)).

However, image roughness failed to characterize the extent of hair surface damage in SEM images, as the metric does not sufficiently correlate surface roughness with variation in pixel brightness. Average image roughness, optimized with summation of brightness differences in 100 sections, yielded a correlation of only 0.259 with SEM damage grade [12] ($p = 0.153$; figure 3$b$), after evaluation over a range of 10, 20, 50, 100, and pixel-by-pixel sections. For example, two images exhibited similar average image roughness, as calculated using equations (2.7)–(2.9), despite showing substantially different extents of surface damage, assessed using the SEM damage grade system; the mostly undamaged hair was even associated with a greater average roughness than one displaying holes in the cuticle and partial exposure of the cortex, likely from overrepresentation of overlapped cuticles in the former (electronic supplementary material, figure S1b) and underrepresentation of holes and partial cortex exposure in the latter image (electronic supplementary material, figure S1a). It is obvious that average image roughness does not adequately represent morphological features of damage, and thus, is not an appropriate metric for scoring hair surface damage.

Instead, analysis of pixel brightness histograms in a manner similar to chromatographic peak tailing more effectively captured roughness associated with hair surface damage. Average pixel brightness histograms showed pronounced peak lag tailing for SEM images of exploded hairs compared to controls (figure 3$c$), linked to the pixel brightness of damage features, and thus, to roughness. As damage features accumulate in an image, surface roughness increases along with a higher proportion of brighter pixels, thereby positively skewing the image brightness profile and creating a tailing effect.

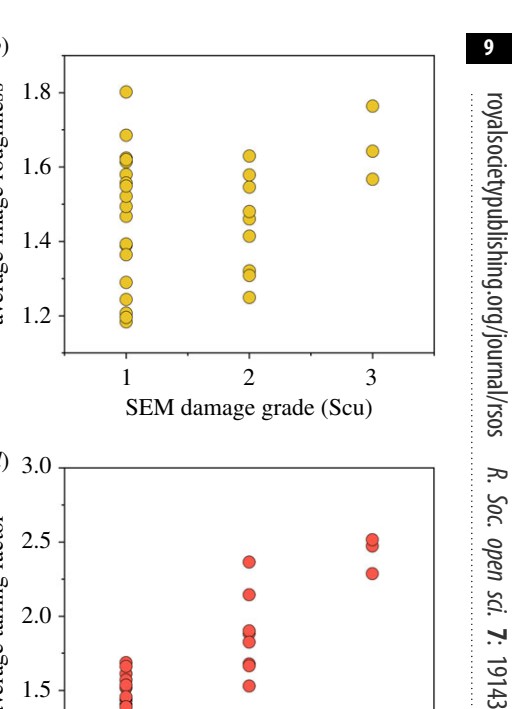

**Figure 3.** Image metrics and parameters for characterization of hair surface damage in SEM images. (*a*) Correlation between average image brightness and magnification after normalization. A moderate correlation (Pearson product-moment correlation (PPMC) coefficient $r = 0.595$, $p = 8.26 \times 10^{-7}$, d.f. $= 56$) between image brightness and magnification indicates that images acquired with vastly different magnifications cannot be compared without an alternative normalization scheme. Thus, only images with magnification less than or equal to 4000× were considered for damage scoring. (*b*) Correlation of average image roughness, as calculated using equations (2.7)–(2.9), with SEM damage grade (PPMC coefficient $r = 0.259$, $p = 0.153$, d.f. $= 30$). Roughness was calculated for 100 sections along the horizontal axis. As evidenced by the wide range of roughness measurements for images designated as sustaining Scu 1 damage, average image roughness does not sufficiently represent hair surface damage. (*c*) Average image brightness histograms for exploded and control SEM hair images with inset. Inset shows pronounced peak tailing in histograms of exploded hairs compared to control hair image brightness histograms, which can be further exploited to describe hair damage. (*d*) Correlation of tailing factor with SEM damage grade (PPMC coefficient $r = 0.823$, $p = 7.31 \times 10^{-9}$, d.f. $= 30$). Tailing factor, a measure of peak tailing, was calculated at 2% of the peak height maximum. Compared to image roughness, tailing factor better captures the extent of hair surface damage in SEM images.

Tailing factor, conventionally used to characterize peak shape in chromatographic separations [31], was investigated as a metric to represent tailing in a pixel brightness histogram and characterize roughness from hair surface damage. As the ratio of the full peak width to twice the peak lead width, typically calculated at 5% of the peak height maximum in chromatography [31], a tailing factor of 1 indicates a symmetrical peak and thus, the absence of tailing, while values greater than 1 indicate peak lag tailing. For quantification of hair surface roughness, the tailing factor for pixel brightness histograms yielded maximum statistically significant correlation with SEM damage grade when determined at 2% of the height maximum ($r = 0.823$; $p = 7.31 \times 10^{-9}$; figure 3*d*); the metric was optimized between 1 and 10% of the peak height maximum. In contrast to average image roughness, comparison of SEM images in electronic supplementary material, figure S1a and S1b demonstrated good agreement between SEM damage grade and tailing factor; small holes, lifting of the cuticle edges and peeling of a cuticle layer partially exposing the cortex were features in the exploded hair that contributed to a tailing factor of 2.473, compared to a tailing factor of 1.451 in electronic supplementary material, figure S1b for a control hair.

Tailing factor further represents more generalized features of hair surface damage and requires no predefined characteristics for scoring of damage. Given the strong correlation, a kNN model was developed and tested for prediction of SEM damage grade using tailing factor; with three nearest neighbours, 81% classification accuracy was achieved (table 1), reiterating the success of capturing the

**Table 1.** Predicted microscopy damage grade and probability of prediction for SEM hair images in test set from kNN model with $k = 3$ based on tailing factor calculated at 2% of peak height maximum.

| hair sample | sample damage classification | SEM damage grade | tailing factor | predicted damage grade | probability |
|---|---|---|---|---|---|
| 4 | control | Scu 1 | 1.182 | Scu 1 | 1 |
| 3 | exploded | Scu 1 | 1.197 | Scu 1 | 1 |
| 4 | control | Scu 1 | 1.209 | Scu 1 | 1 |
| 5 | control | Scu 2 | 1.269 | Scu 1[a] | 1 |
| 3 | exploded | Scu 1 | 1.306 | Scu 1 | 1 |
| 3 | exploded | Scu 1 | 1.403 | Scu 1 | 1 |
| 5 | control | Scu 1 | 1.434 | Scu 1 | 1 |
| 4 | control | Scu 1 | 1.451 | Scu 1 | 1 |
| 2 | exploded | Scu 1 | 1.521 | Scu 1 | 0.667 |
| 1 | exploded | Scu 1 | 1.570 | Scu 1 | 0.667 |
| 2 | exploded | Scu 1 | 1.687 | Scu 2[a] | 0.667 |
| 3 | exploded | Scu 2 | 1.825 | Scu 2 | 0.667 |
| 1 | exploded | Scu 2 | 1.883 | Scu 2 | 0.667 |
| 1 | exploded | Scu 2 | 1.901 | Scu 2 | 0.667 |
| 1 | exploded | Scu 2 | 2.365 | Scu 3[a] | 0.667 |
| 1 | exploded | Scu 3 | 2.473 | Scu 3 | 0.667 |

[a]Incorrectly predicted damage grade.

same features defined by the SEM damage grade system. However, three misclassified images highlight the limitations of a classification system based on specific microscopic features. For example, a higher damage grade was predicted for electronic supplementary material, figure S2a, an exploded hair, initially classified as having Scu 2 damage from lift-up of the cuticle and presence of holes. But the presence of embedded particulates and residue remaining after removal from adhesive were ignored as they were not specified features in the damage grade criteria, though these features contribute prominently to surface roughness and damage in the image. On the other hand, tailing factor enabled prediction of a relatively undamaged control hair to Scu 1, though classified as sustaining Scu 2 damage due to the presence of a hole and a few cuticle lift-ups (electronic supplementary material, figure S2b). Classification systems based on presence or absence of defined features do not provide quantitative scoring for images based on extent of damage. Tailing factor overcomes limitations of hair damage classification systems, as it is intrinsically linked to the magnitude of surface damage and it enables successful scoring of images without prior identification of specific features.

Applied to each hair specimen, average tailing factor and its range across images describe hair damage severity more completely. For example, compared to Hair Sample 1, tailing factors for images of different regions along Hair Sample 3 are smaller (table 1), thus indicating less severe cuticular damage even though both are exploded hairs. Indeed, some tailing factors for Hair Sample 3 images in the test set are similar to those for Hair Sample 4, an undamaged control hair. However, when accounting for all of the tailing factors from SEM images of different regions along Hair Sample 3, including those from the training set, a larger average tailing factor is attained (1.545 ± 0.363 (s.d.)), with a wider tailing factor range (minimum = 1.197, maximum = 2.145), as compared to Hair Sample 4 (1.337 ± 0.153; minimum = 1.182, maximum = 1.535). Clearly, more damage has been sustained by Hair Sample 3, given the larger average tailing factor. The wider tailing factor range in the exploded hair further indicates the presence of both damaged and undamaged regions, signifying non-uniform severity of cuticular damage along the hair, whereas Hair Sample 4 is primarily undamaged. Collectively, the magnitude and range of tailing factors for each hair quantify the severity of cuticular damage in a more comprehensive manner. Not limited to hair analysis, this novel quantitative metric may be applied widely for assessments of surface damage in other materials relevant to the life and material sciences.

# 4. Conclusion

We offer a quantitative and objective approach to assess hair surface damage from scanning electron microscopic images. As a proxy for surface roughness, tailing factor quantifies the severity of hair cuticular damage by exploiting pixel brightness in elevated and depressed morphological features of damage. Successful characterization of morphological features unique to exploded hairs further demonstrates the ability for tailing factor to accommodate a diverse set of features as a broad metric to probe surface topography, which enables an investigation into the correlations of morphological damage and chemical composition changes in exploded hairs, and may find utility in disciplines such as medical, forensic, and material sciences to provide quantitative microscopic analyses of mechanical damage in hair and other materials.

Ethics. Hair specimens were collected from individuals under approval by the Institutional Review Board at Lawrence Livermore National Laboratory (Protocol ID# 15-008) and in accordance with the Declaration of Helsinki. Written informed consent for specimen collection and analysis was obtained prior to collection.

Data accessibility. Code for the normalization procedure and tailing factor calculation to be performed in ImageJ has been included in electronic supplementary material as appendix S1. Code for calculation of average image roughness has been included in electronic supplementary material as appendix S2. Additional scanning electron microscopic images have been deposited into the Dryad Data Repository: https://dx.doi.org/10.5061/dryad.ttdz08kt4.

Authors' contributions. F.C. participated in the study design, prepared the samples, acquired and analysed the data, and drafted the manuscript; D.S.A. conceived of the study, designed the study, performed the sample collection, and revised the manuscript; A.D.J. participated in the study design and revised the manuscript; B.R.H. coordinated the study and participated in the study design. All authors gave final approval for publication and agree to be held accountable for the work performed therein.

Competing interests. We declare we have no competing interests.

Funding. This work was financially supported by the Department of Energy, Lawrence Livermore National Laboratory, Laboratory Directed Research and Development award (16-SI-002). F.C. acknowledges support from the Lawrence Livermore National Laboratory Livermore Graduate Scholars Program. This work was performed under the auspices of the U.S. Department of Energy by Lawrence Livermore National Laboratory under Contract DE-AC52-07NA27344. A.D.J. acknowledges support from Michigan AgBioResearch through the USDA National Institute of Food and Agriculture, Hatch project number MICL02474.

Acknowledgements. The authors thank Dr Zurong Dai (Lawrence Livermore National Laboratory) for use of instrumentation and assistance in acquiring scanning electron micrographs, and the ATF National Center for Explosives Training and Research at the Redstone Arsenal (Huntsville, AL) for hosting the training exercise.

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
