## [Reviewer comments · Royal Society Open Science]

Review History

RSOS-191438.R0 (Original submission)

Review form: Reviewer 1

Is the manuscript scientifically sound in its present form?

Yes

Are the interpretations and conclusions justified by the results?

Yes

Is the language acceptable?

Yes

Do you have any ethical concerns with this paper?

No

Have you any concerns about statistical analyses in this paper?

No

Recommendation?

Accept as is

Comments to the Author(s)

The paper is novel in developing an automated method for hair surface analysis using SEM. The authors' responses to the reviewers are acceptable.

Review form: Reviewer 2

Is the manuscript scientifically sound in its present form?

No

Are the interpretations and conclusions justified by the results?

No

Is the language acceptable?

Yes

Do you have any ethical concerns with this paper?

No

Have you any concerns about statistical analyses in this paper?

No

Recommendation?

Reject

Comments to the Author(s)

This manuscript deals with hair observation by SEM to classify the damage associated from i.e. blast explosions. This manuscript is a transfer from a previous journal. Experiments are well-conducted but, in my opinion, there is not any novelty here. Hair damage can also come from the use of a hairdryer or dyes to change the color. Why to use a very expensive technique like SEM? I agree with one of the reviewers of the previous submission, DNA analysis is more suitable. As for the experimental part, is well-conducted, yet is merely routine analysis without any other novelty. As such, I cannot recommend publication in Royal Society Open Science.

Review form: Reviewer 3

Is the manuscript scientifically sound in its present form?

Yes

Are the interpretations and conclusions justified by the results?

Yes

Is the language acceptable?

Yes

Do you have any ethical concerns with this paper?

No

Have you any concerns about statistical analyses in this paper?

No

Recommendation?

Accept with minor revision (please list in comments)

Comments to the Author(s)

This paper presents that aimed to develop and apply a quantitative measure for scoring hair surface damage in scanning electron microscopic (SEM) images without predefined features and automate image analysis for characterization of morphological hair damage after exposure to an explosive blast. Assessment of hair damage with tailing factor, a measure of asymmetry in pixel brightness histograms and proxy for surface roughness, yielded 81% classification accuracy to an existing damage classification system, indicating good agreement between the two metrics. The article is of interest and innovation. I think the article can be accepted after minor changes.

(1) It is suggested to revise the abstract to highlight the innovation points.

(2) The introduction of recent progress of novel semiconducting materials in electronics and optoelectronics may attract broader readership. For example, DOI: 10.1166/sam.2019.3487; 10.20964/2017.09.06; 10.1002/adma.201705421;

10.1002/ADOM.201700984; 10.1016/j.jpowsour.2019.227149; 10.1039/c5ra09405d; 10.1007/s11705-018-1754-3; 10.1021/acsnano.8b08079;

(3) Figure captions should be more informative.

Decision letter (RSOS-191438.R0)

15-Oct-2019

Dear Ms Chu:

Title: Automated analysis of scanning electron microscopic images for assessment of hair surface damage

Manuscript ID: RSOS-191438

The editor assigned to your manuscript has now received comments from reviewers. We would like you to revise your paper in accordance with the referee and Subject Editor suggestions which can be found below (not including confidential reports to the Editor). Please note this decision does not guarantee eventual acceptance.

Please submit your revised paper before 07-Nov-2019. Please note that the revision deadline will expire at 00.00am on this date. If we do not hear from you within this time then it will be assumed that the paper has been withdrawn. In exceptional circumstances, extensions may be possible if agreed with the Editorial Office in advance. We do not allow multiple rounds of revision so we urge you to make every effort to fully address all of the comments at this stage. If

deemed necessary by the Editors, your manuscript will be sent back to one or more of the original reviewers for assessment. If the original reviewers are not available we may invite new reviewers.

Please also include the following statements alongside the other end statements. As we cannot publish your manuscript without these end statements included, if you feel that a given heading is not relevant to your paper, please nevertheless include the heading and explicitly state that it is not relevant to your work.

- Acknowledgements

RSC Associate Editor:

Comments to the Author:

According to the comments of the adjudicator, the decision was made.

RSC Subject Editor:

Comments to the Author:

(There are no comments.)

Reviewers' Comments to Author:

Reviewer: 1

Comments to the Author(s)

The paper is novel in developing an automated method for hair surface analysis using SEM. The authors' responses to the reviewers are acceptable.

Reviewer: 2

Comments to the Author(s)

This manuscript deals with hair observation by SEM to classify the damage associated from i.e. blast explosions. This manuscript is a transfer from a previous journal. Experiments are well-conducted but, in my opinion, there is not any novelty here. Hair damage can also come from the use of a hairdryer or dyes to change the color. Why to use a very expensive technique like SEM? I agree with one of the reviewers of the previous submission, DNA analysis is more suitable. As for the experimental part, is well-conducted, yet is merely routine analysis without any other novelty. As such, I cannot recommend publication in Royal Society Open Science.

Reviewer: 3

Comments to the Author(s)

This paper presents that aimed to develop and apply a quantitative measure for scoring hair surface damage in scanning electron microscopic (SEM) images without predefined features and automate image analysis for characterization of morphological hair damage after exposure to an explosive blast. Assessment of hair damage with tailing factor, a measure of asymmetry in pixel brightness histograms and proxy for surface roughness, yielded 81% classification accuracy to an existing damage classification system, indicating good agreement between the two metrics. The article is of interest and innovation. I think the article can be accepted after minor changes.

(1) It is suggested to revise the abstract to highlight the innovation points.

(2) The introduction of recent progress of novel semiconducting materials in electronics and optoelectronics may attract broader readership. For example, DOI: 10.1166/sam.2019.3487; 10.20964/2017.09.06; 10.1002/adma.201705421;

10.1002/ADOM.201700984; 10.1016/j.jpowsour.2019.227149; 10.1039/c5ra09405d; 10.1007/s11705-018-1754-3; 10.1021/acsnano.8b08079;

(3) Figure captions should be more informative.

Author's Response to Decision Letter for (RSOS-191438.R0)

See Appendix A.

Decision letter (RSOS-191438.R1)

19-Nov-2019

Dear Ms Chu:

Title: Automated analysis of scanning electron microscopic images for assessment of hair surface damage

Manuscript ID: RSOS-191438.R1

It is a pleasure to accept your manuscript in its current form for publication in Royal Society Open Science. The chemistry content of Royal Society Open Science is published in collaboration with the Royal Society of Chemistry.

RSC Associate Editor
Comments to the Author:
(There are no comments.)

Reviewer(s)' Comments to Author:

Appendix A

Manuscript ID RSOS-191438

Title: Automated analysis of scanning electron microscopic images for assessment of hair surface damage

The section “Acknowledgements” immediately follows the “Conclusions” and “Data Availability” sections in the manuscript.

Responses to reviewers' comments

Reviewers' Comments to Author:

Reviewer: 1

Comments to the Author(s)

The paper is novel in developing an automated method for hair surface analysis using SEM. The authors' responses to the reviewers are acceptable.

We thank the reviewer for the kind comment.

Reviewer: 2

Comments to the Author(s)

This manuscript deals with hair observation by SEM to classify the damage associated from i.e. blast explosions. This manuscript is a transfer from a previous journal. Experiments are well-conducted but, in my opinion, there is not any novelty here. Hair damage can also come from the use of a hairdryer or dyes to change the color. Why to use a very expensive technique like SEM? I agree with one of the reviewers of the previous submission, DNA analysis is more suitable. As for the experimental part, is well-conducted, yet is merely routine analysis without any other novelty. As such, I cannot recommend publication in Royal Society Open Science.

The authors respectfully disagree with the reviewer's critique that this manuscript offers no novelty. This paper describes a novel approach to microscopic analysis, primarily in automating image analysis and interpretation after scanning electron microscopic image acquisition to examine damage to hair surfaces. As such, while experimental SEM image acquisition of hair fibers remains routine, the innovation lies in the development of an automated approach to quantify observed hair damage in SEM images. Where conventionally, microscopic analysis, including SEM, has been utilized for qualitative, often subjective, analyses, as discussed in the Introduction of this paper, this manuscript provides an objective method to address the question: How damaged is “damaged”?

We recognize and agree that mechanical hair damage can arise from other processes, such as hair dryer use and hair dyes. Although this automated method for hair damage assessment was developed by comparing hairs that have been recovered after an explosive blast, tailing factor can be broadly applicable to quantify hair damage from various mechanisms, including hair dryer use and hair dyes, as the metric does not require user input or selection of defined features indicative of damage. Indeed, some of the damage features from hair dryer use are observed in exploded hairs, as both the explosive blast and hair-drying result in thermal damage. This point

was discussed in the first paragraph of the subsection “Identification of microscopic features for characterization of hair surface damage” in the Results and Discussion section of the manuscript. Broad applicability of our metric to quantify the severity of hair surface damage was emphasized at the end of the Results and Discussion section and in the Conclusions section of the manuscript.

Regarding the justification for using an expensive technique such as SEM to assess hair damage, this technique and even transmission electron microscopy (TEM) are regularly employed to observe fine details that are not evident using light microscopy and that are associated with symptoms of diseases affecting hair, such as alopecia, lamellar ichthyosis, and trichothiodystrophy (Rice et al. (1996) [5]), and to evaluate the efficacy of hair care products towards hair damage recovery (Kim et al. (2010) [12]). Both articles are cited in the Introduction section of this manuscript.

We believe that the reviewer has misunderstood the critique provided by a reviewer in the previous submission regarding DNA analysis and its suitability for characterizing hair damage, though as an aside, genomic DNA is often not present in hair, which makes DNA analysis in hair unreliable. Below is the relevant excerpt of the critique provided by a previous reviewer:

“In the broader sense SEM in the forensic examination of hairs has not proven to be especially useful . Although there are criticisms of microscopic examination this remains the foundation of forensic practice with the emphasis on selecting hairs for DNA analysis. DNA analysis does not substitute for the criminalistics information , such as has a hair been exposed to explosives etc . Hence, the paper does have some merit albeit limited by the very small sample size.”

In the context of hair damage assessment, DNA analysis would not yield relevant information; the purpose of examining hair fibers for damage is to determine the extent of hair damage, such as for clinical diagnoses, or for other criminalistics information (e.g., has the hair been exposed to explosives), as stated by a previous reviewer, and not for human identification purposes, which is the forte of DNA analyses.

In addition to automating image analysis to quantify the severity of hair surface damage and even beyond hair to other materials, this work aims to support an investigation into whether microscopic hair analysis can be used as a predictor of proteomic profiling success, as described in the last paragraph of the Introduction and also in the Conclusions section of this manuscript. Through the development and application of our metric, tailing factor, for hair damage assessment, which has not previously been exploited in this manner, morphological hair surface damage can now be quantified and compared against changes in hair chemical composition from an explosive blast.

Reviewer: 3

Comments to the Author(s)

This paper presents that aimed to develop and apply a quantitative measure for scoring hair surface damage in scanning electron microscopic (SEM) images without predefined features and automate image analysis for characterization of morphological hair damage after exposure to an explosive blast. Assessment of hair damage with tailing factor, a measure of asymmetry in pixel brightness histograms and proxy for surface roughness, yielded 81% classification accuracy to an existing damage classification

system, indicating good agreement between the two metrics. The article is of interest and innovation. I think the article can be accepted after minor changes.

The authors thank the reviewer for the kind comment.

(1) It is suggested to revise the abstract to highlight the innovation points.

The second sentence of the abstract has been revised to the following to emphasize the novelty of this work:

“We describe the development and application of a novel quantitative measure for scoring hair surface damage in scanning electron microscopic (SEM) images without predefined features, and automation of image analysis for characterization of morphological hair damage after exposure to an explosive blast.”

(2) The introduction of recent progress of novel semiconducting materials in electronics and optoelectronics may attract broader readership. For example, DOI: 10.1166/sam.2019.3487; 10.20964/2017.09.06; 10.1002/adma.201705421; 10.1002/ADOM.201700984; 10.1016/j.jpowsour.2019.227149; 10.1039/c5ra09405d; 10.1007/s11705-018-1754-3; 10.1021/acsnano.8b08079;

Below are the papers to which the reviewer refers:

10.1166/sam.2019.3487

Structural Design and Electrochemical Performance of PANI/CNTs and MnO₂/CNTs Supercapacitor

Guo-Ting Xia, Chen Li, Kai Wang, Li-Wei Li (2019)

10.20964/2017.09.06

Electrodeposition Synthesis of PANI/MnO₂/Graphene Composite Materials and its Electrochemical Performance

Kai Wang, Liwei Li, Wen Xue, Shengzhe Zhou, Yong Lan, Hongwei Zhang, Zongqiang Sui (2017)

10.1002/adma.201705421

VO₂/TiN Plasmonic Thermochromic Smart Coatings for Room-Temperature Applications

Qi Hao, Wan Li, Huiyan Xu, Jiawei Wang, Yin Yin, Huaiyu Wang, Libo Ma, Fei Ma, Xuchuan Jiang, Oliver G. Schmidt, Paul K. Chu (2018)

10.1002/ADOM.201700984

Boosting the Photoluminescence of Monolayer MoS₂ on High-Density Nanodimer Arrays with Sub-10 nm Gap

Qi Hao, Jinbo Pang, Yang Zhang, Jiawei Wang, Libo Ma, Oliver G. Schmidt (2017)

10.1016/j.jpowsour.2019.227149

Remaining useful life prediction for supercapacitor based on long short-term memory neural network

Yanting Zhou, Yinuo Huang, Jinbo Pang, Kai Wang (2019)

10.1039/c5ra09405d

Direct synthesis of graphene from adsorbed organic solvent molecules over copper
Jinbo Pang, Alicja Bachmatiuk, Lei Fu, Rafael G. Mendes, Marcin Libera, Daniela Placha, Grazyna Simha Martynková, Barbara Trzebicka, Thomas Gemming, Juergen Eckert, Mark H. Rümmeli (2015)

10.1007/s11705-018-1754-3

A free-standing superhydrophobic film for highly efficient removal of water from turbine oil
Fan Shu, Meng Wang, Jinbo Pang, Ping Yu (2019)

10.1021/acsnano.8b08079

Electron-Driven *In Situ* Transmission Electron Microscopy of 2D Transition Metal Dichalcogenides and Their 2D Heterostructures

Rafael G. Mendes, Jinbo Pang, Alicja Bachmatiuk, Huy Quang Ta, Liang Zhao, Thomas Gemming, Lei Fu, Zhongfan Liu, Mark H. Rümmeli (2019)

The articles suggested by the reviewer focus on semiconductor materials, whereas our work aims to develop an objective method for characterizing hair surface damage, and as such, has concentrated on demonstrating the gaps in microscopic hair analysis and interpretation from this approach, one of which is in the qualitative nature of SEM analysis of hair. We believe that a thorough discussion of semiconductor materials is outside the scope of this manuscript. However, an additional benefit of our metric for quantifying hair surface damage is that it can be broadly applied to other materials, such as the semiconductor materials suggested by the reviewer. Thus, we have included the potential for applicability of characterizing surface damage of other materials, such as semiconductor materials, at the end of the Results and Discussion section and also in the Conclusions section, respectively, as follows:

“Not limited to hair analysis, this novel quantitative metric may be applied widely for assessments of surface damage in other materials relevant to the life and material sciences.”

“Successful characterization of morphological features unique to exploded hairs further demonstrates the ability for tailing factor to accommodate a diverse set of features as a broad metric to probe surface topography, which enables an investigation into the correlations of morphological damage and chemical composition changes in exploded hairs, and may find utility in disciplines such as medical, forensic, and **material** sciences to provide quantitative microscopic analyses of mechanical damage in hair and other materials.”

(3) Figure captions should be more informative.

In lieu of any specific suggestions on how the figure captions can be improved, the following figure captions have been revised, with additions in red:

Figure 1. SEM images of (a) and (b) hair segments as original raw images, (c) and (d) rotated segments with selected region of interest (yellow rectangle) and image brightness histogram of

region, and (e) and (f) regions of interest after normalization and corresponding image brightness histograms, from Hair Samples 4 and 2, respectively. Original images (c) and (d) show different illumination levels due to hair-to-hair variation and image acquisition differences. The described normalization procedure minimized brightness differences within and between images, as displayed in (e) and (f).

Figure 2. Representative rotated SEM images with overlays of normalized regions of interest and corresponding brightness histograms from Hair Samples 1 – 5, respectively. Features are labeled and denoted by yellow arrows. In addition to debris and particulates on the hair surface, features characteristic of damage from an explosion induced by an explosive device include (a) holes exposing layers of cuticle, (b) severe lifting of the cuticle edges and large cracks leading to partial exposure of cortex, and (c) localized non-specific cuticle lifting with residue from adhesive tape. Undamaged control hairs (d) and (e) predominantly display overlapped cuticles from daily weathering, illustrating substantially less severe hair surface damage compared to exploded hairs.

Figure 3. Image metrics and parameters for characterization of hair surface damage in SEM images. (a) Correlation between average image brightness and magnification after normalization. A moderate correlation (Pearson product-moment correlation (PPMC) coefficient $r = 0.595$, $p = 8.26 \times 10^{-7}$, $df = 56$) between image brightness and magnification indicates that images acquired with vastly different magnifications cannot be compared without an alternative normalization scheme. Thus, only images with magnification $\leq 4000X$ were considered for damage scoring. (b) Correlation of average image roughness, as calculated using Equations 7 – 9, with SEM damage grade (PPMC coefficient $r = 0.259$, $p = 0.153$, $df = 30$). Roughness was calculated for 100 sections along the horizontal axis. As evidenced by the wide range of roughness measurements for images designated as sustaining Scu 1 damage, average image roughness does not sufficiently represent hair surface damage. (c) Average image brightness histograms for exploded and control SEM hair images with inset. Inset shows pronounced peak tailing in histograms of exploded hairs compared to control hair image brightness histograms, which can be further exploited to describe hair damage. (d) Correlation of tailing factor with SEM damage grade (PPMC coefficient $r = 0.823$, $p = 7.31 \times 10^{-9}$, $df = 30$). Tailing factor, a measure of peak tailing, was calculated at 2% of the peak height maximum. Compared to image roughness, tailing factor better captures the extent of hair surface damage in SEM images.